# Oscillatory Behavior of Pd-Au Catalysts in Toluene Total Oxidation

**Tarek Barakat [1,2,*]**, **Joanna C. Rooke [1]**, **Dayan Chlala [3]**, **Renaud Cousin [2]**,
**Jean-François Lamonier [3]**, **Jean-Marc Giraudon [3]**, **Sandra Casale [4]**, **Pascale Massiani [4]**,
**Bao-Lian Su [1,*]** **and Stéphane Siffert [2,*]**

[1] Laboratory of Inorganic Materials Chemistry, University of Namur, 61 Rue de Bruxelles, 5000 Namur, Belgium; jc_rooke@hotmail.com

[2] Unité de Chimie Environnementale et Interactions sur le Vivant (UCEIV) E.A.4492, Université du Littoral Côte d'Opale, 145 Avenue Maurice Schumann, 59140 Dunkerque, France; Renaud.Cousin@univ-littoral.fr

[3] Univ. Lille, CNRS, Centrale Lille, ENSCL, Univ. Artois, UMR 8181-UCCS-Unité de Catalyse et Chimie du Solide, 59000 Lille, France; dayanechlela@hotmail.com (D.C.); jean-francois.lamonier@univ-lille1.fr (J.-F.L.); jean-marc.giraudon@univ-lille1.fr (J.-M.G.)

[4] UMR CNRS 7197—UPMC, 4 Place Jussieu, Casier 178, 75005 Paris, France; sandra.casale@upmc.fr (S.C.); pascale.massiani@upmc.fr (P.M.)

**\*** Correspondence: tarek.barakat@unamur.be (T.B.); bao-lian.su@unamur.be (B.-L.S.); siffert@univ-littoral.fr (S.S.); Tel.: +32-81-725-413 (T.B.); +32-81-724-531 (B.-L.S.); +33-328-658-256 (S.S.)

**Abstract:** In this work, the activity of bimetallic Pd-Au doped hierarchically structured titania catalysts has been investigated in the total oxidation of toluene. In earlier works, doping titania with group Vb metal oxides ensured an increased catalytic performance in the elimination of VOC molecules. A synergy between gold and palladium loaded at the surface of titania supports provided better performances in VOC oxidation reactions. Therefore, the main focus in this work was to investigate the durability of the prepared catalysts under long time-on-stream periods. Vanadium-doped catalysts showed a stable activity throughout the whole 110 h test, whereas, surprisingly, niobium-doped catalysts presented a cycle-like activity while nevertheless maintaining a high performance in toluene elimination. Operando Diffuse Reflectance Infrared Fourrier Transform spectroscopy (DRIFT) experiments revealed that variations in the presence of OH radicals and the presence of carbonaceous compounds adsorbed at the surface of spent catalysts varies with the occurrence of oscillations. X-ray Photoelectron Spectroscopy (XPS) results show that interactions between the material and the active phase provided extra amounts of mobile oxygen species and participated in easing the reduction of palladium. An enhanced redox reaction scheme is thus obtained and allows the occurrence of the cyclic-like performance of the catalyst.

**Keywords:** oscillatory behavior; bimetallic catalysts; hierarchically porous; doping; titania; palladium; gold; VOC oxidation

## 1. Introduction

The total oxidation of volatile organic compounds over noble metal-based catalysts has been widely studied in the past couple of decades to reduce atmospheric pollution produced by industrial emissions. Reviews discussing the use of palladium and gold-based mono or bimetallic catalytic systems for volatile organic compounds' (VOC) removal or water-gas shit reactions can be easily found in the literature. In fact, palladium and gold have been investigated in the oxidation of many VOC molecules, such as propene [1,2], toluene [3–5], chlorobenzene [6], ethanol [7], butanone [8], etc. It has been found that loading both Pd and Au on the same support enhances the catalyst performance

compared to that of monometallic catalysts. For example, Enache et al. [9,10] investigated gold and palladium catalysts supported on $TiO_2$ in the oxidation of alcohols and aldehydes, and showed the effect of a synergy between both noble metals, leading to an increased activity and to a greater control of the selectivity in an oxidation reaction. Hosseini et al. [11–13] also studied Pd-Au catalysts in the total oxidation of toluene and propene, and they concluded that these catalysts possess higher activities than monometallic Au or Pd. They also proved that a gold-rich core and shell-rich palladium morphology provides the best results in VOC elimination in terms of activity.

As for the catalytic elimination, different oxidation modes of pollutant molecules have been studied. In most cases, a linear conversion pattern of pollutant molecules is observed, whereas in some cases, oscillatory behaviours occurred. For example, research studies in the field of methane total oxidation show an oscillatory behavior that may occur naturally under methane-rich reactant flow mixtures. Graham et al. [14] first observed this oscillatory behaviour during methane oxidation over a thick Pd-film catalyst. Deng et al. [15] and Zhang et al. [16] also reported a periodical oscillatory conversion of methane using Pd-supported and metallic Pd foil catalysts, respectively. Both works attribute these oscillations to the use of palladium as an active phase constituent. Deng et al. also reported that adding small amounts of ceria to the oxide support could help stabilise the oscillations. Other works by Zhang et al. [17,18] concluded that oscillations may also be observed when using metallic nickel or cobalt foils. However, their intensities and frequencies were lower than what had been observed with palladium [16]. Furthermore, Bychkov et al. [19–21] confirmed the latter results through witnessing self-oscillations on Pd-supported and Ni-supported catalysts as well as metal foils (Ni and Co). They also reported this phenomenon in the oxidation of ethane under an alkane-rich mixture. Variations in surface oxygen amounts helped verify that these oscillations are due to the redox changes of the active phase constituents. Lashina et al. [22] discussed self-sustained oscillations occurring in the oxidation of CO over $PdO/Al_2O_3$ at stable frequencies of 40 s and also correlated them with changes in the oxidation state of Pd particles as well as surface changes of the catalyst induced by oxygen. Lee et al. [23] observed oscillations in the oxidation of toluene on a NaX-type zeolite at a rate of 42 to 48 min. They claimed that the successive deactivation/regeneration cycles of the catalyst were responsible for its cycle-like performance. However, in all these studies, observed oscillations occurred periodically at time intervals of a maximum of 50 min.

In this work, an unprecedented oscillatory behaviour has been detected in the oxidation of toluene over bimetallic Pd-Au-loaded hierarchically structured doped titania catalysts. Therefore, the present work focuses on understanding the kinetic oscillations occurring on these catalysts under exposure of a toluene-air mixture for 100 h by using operando DRIFT spectroscopy.

## 2. Results

PdAu5NbTi and PdAu5VTi catalysts previously tested in the total oxidation of toluene in increasing temperature were exposed to 110 h of a flowing toluene (1000 ppm) and air mixture at a low conversion rate (~15% conversion to $CO_2$, T15(PdAu5NbTi) = 200 °C and T15(PdAu5VTi) = 212 °C). At a 15% conversion rate, an enhanced deactivation process is often observed compared to that seen at a 100% conversion rate, at which deactivation is also more difficult to observe [13,24,25]. This deactivation can be caused by the formation and deposition of greater amounts of carbonaceous compounds induced by more incomplete combustion at this low conversion rate. Textural characteristics and noble metal content of these samples are shown in Table S1. Figure 1 shows the evolution of toluene conversion to $CO_2$ during the time of exposure to the toluene/air mixture. The PdAu5VTi catalyst showed a stable conversion rate of 15% throughout the whole 110 h test (Figure 1a), whereas toluene conversion over PdAu5NbTi (Figure 1b) appeared to follow a cyclic pattern. Two types of oscillations are clearly seen in Figure 1b: The first consists of a low catalytic performance with small variation between 13 and 16% at two time intervals (10–30 h and ~70–100 h), whereas the second represents a significantly higher catalytic performance where a variation reaches approximately 21% at 40 h and falls sharply back to around 15% at around 70 h.

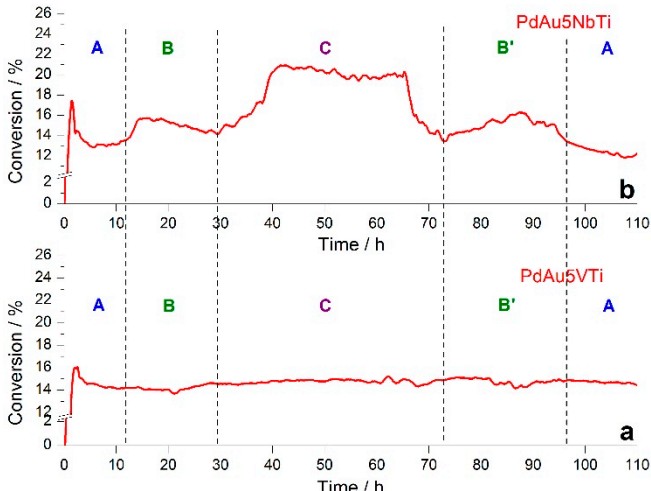

**Figure 1.** Toluene conversion rate versus time in the ageing of (**a**) PdAu5VTi and (**b**) PdAu5NbTi catalysts.

In the literature, Tsou et al. observed an oscillating effect during the oxidation of methylisobutylketone (MIBK) over a Pt/HFAU catalyst at fixed temperatures for 3 h [26,27]. They claimed that the oscillating effect is caused by the oxidation of the active coke molecules adsorbed onto the surface and into the pores of the used catalyst. Bychkov et al. [21] also suggested the deposition/desorption of carbonaceous compounds on the catalysts' surfaces is directly related to a self-oscillatory process they investigated in the oxidation of methane and ethane. Along with Deng et al. [15] and Zhang et al. [16], they additionally correlated the oscillatory behaviour to changes in the oxidation state of active phase constituents. Therefore, and for a better understanding of changes occurring at the surface of the catalyst, it was inevitable to conduct operando DRIFT measurements throughout the whole duration of the catalytic test. These experiments would provide insight into changes caused by the constant exposure to pollutants at low conversion rates.

Operando DRIFT experiments were conducted on both catalysts under the same testing conditions. However, for the sake of simplicity, results displayed in Figure 2 correspond only to vibrations observed at the surface of the PdAu5NbTi catalyst during the ageing test. They comprise of measurements taken every 5 h during the whole test duration. Results for the PdAu5VTi sample are provided as supporting information. Table S2 details vibrational bands cited in the text as well as their attributions.

A closer look to the 3100–3500 $cm^{-1}$ region in Figure 2 reveals alternative increases and decreases in the absorption bands' intensity. These variations comply with the pattern of the cyclic performance seen in Figure 1. Absorption bands detected in this region are directly related to the vibration of OH radicals present at the surface of the catalyst. Therefore, the witnessed variations indicate alternative increases and decreases in the presence of these radicals (Table S2). These changes are also accompanied by changes in the shape and position of the peak detected around the $\nu = 3000\ cm^{-1}$ region. Broad and centred at 3070 $cm^{-1}$ at the beginning of the reaction, this peak narrows down and is accompanied by a gradual increase in bands at lower wavenumber values around 3020 $cm^{-1}$. This increase suggests an increasing presence of methyl radicals fragments or originating from linear molecules at the surface of the catalyst whereas at the beginning of the reaction, detected methyl radicals originated mostly from toluene. The following result shows an actual deposition of carbonaceous compounds at the surface of the catalyst. A closer look at results in the range of 1000 to 2000 $cm^{-1}$ shows the appearance of C=O, C-O-H and C-O-C specific absorption bands and a gradual increase in their intensity throughout the testing duration (Table S2). These bands indicate the presence of an interaction between adsorbed compounds and the catalyst surface. They additionally show the presence of oxygenated organic compounds adsorbed at that same surface.

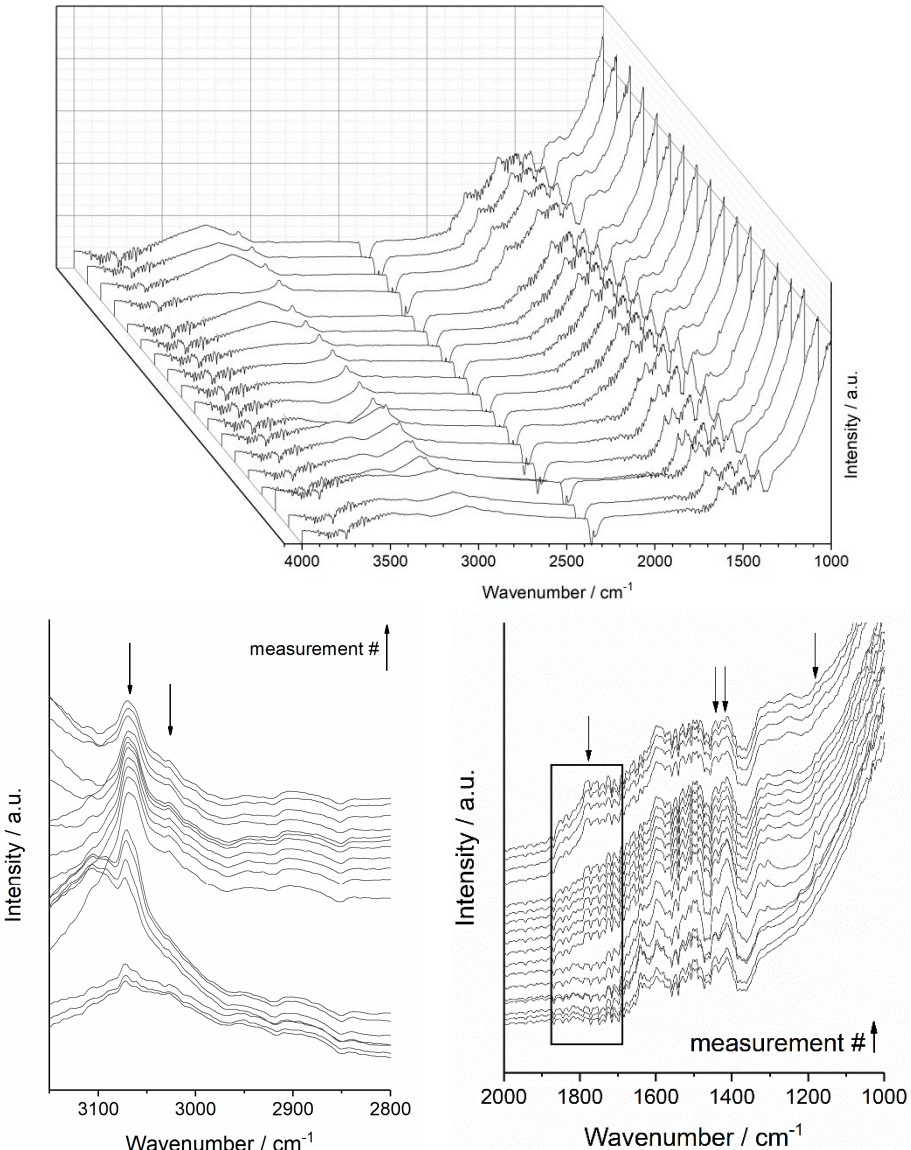

**Figure 2.** Operando DRIFT spectra of PdAu5NbTi catalyst recorded during the ageing test.

XPS experiments were conducted on all doped and undoped, mono and bimetallic samples to evaluate the chemical composition of the catalysts' surfaces. Again, for the sake of clarity, only the O1s XPS spectra of niobium-doped samples will be presented, the remaining results are provided as supporting information. Spectra shown in Figure 3 and values calculated in Table 1 show an increase in atomic oxygen yields after doping with niobium. Peak positions reveal two types of oxygen species: $O_I$ at low BE values, assigned to lattice oxygen ($O^{2-}$), and $O_{II}$, at high BE values, ascribed to surface adsorbed or chemisorbed oxygen ($O_2^-$ or $O^-$), -OH groups, and oxygen vacancies. The amount of $O_{II}$ species nearly triples in the PdAu5NbTi catalyst compared to the undoped titania catalyst (Table 1). Therefore, adding niobium enhances the presence of oxygen vacancies. The amount of $O_{II}$ is the highest in the bimetallic catalyst. Therefore, the presence of both gold and niobium enhance the amount of $O_{II}$ species. As for the importance of such high amounts of $O_{II}$, studies by Fronzi et al. [28] and Schaub et al. [29] reveal that the interaction of $O_{II}$ species with molecular oxygen or water provokes the generation of peroxides, superoxides, and hydroxyl species, thus facilitating the dissociation of adsorbates. This claim has been confirmed by Hernandez et al. [30], who explained that an increase in $O_{II}$ species in their Cu-modified cryptomelane oxide rendered the catalyst more efficient in the oxidation of CO.

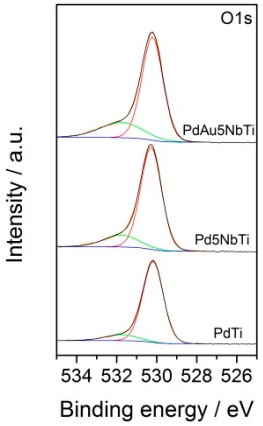

**Figure 3.** O1s XPS spectra of mono and bimetallic Nb-doped TiO$_2$ samples (O$_I$ in brown line, O$_{II}$ in green line).

**Table 1.** Binding energy and quantitative analysis values of XPS experiments conducted on mono and bimetallic Nb-doped catalysts.

| Samples | Binding Energies (eV) | | | | | | | | Quantitative Analysis | | |
|---|---|---|---|---|---|---|---|---|---|---|---|
| | O 1s | | Pd 3d | | Au 4f | | Nb 3d | | O/Ti | O$_I$/(O$_I$ + O$_{II}$) | O$_{II}$/(O$_I$ + O$_{II}$) |
| | O$_I$ | O$_{II}$ | 3d$_{5/2}$ | 3d$_{3/2}$ | 4f$_{7/2}$ | 4f$_{5/2}$ | 2p$_{5/2}$ | 2p$_{3/2}$ | | | |
| PdTi | 530.2 | 531.2 | 336.6 | 341.9 | - | - | - | - | 1.5 | 0.90 | 0.10 |
| Pd5NbTi | 530.3 | 531.8 | 336.7 | 342.0 | - | - | 207.4 | 210.2 | 1.7 | 0.84 | 0.16 |
| PdAu5NbTi | 530.2 | 531.8 | 336.7 | 342.0 | 83.7 | 87.4 | 207.9 | 210.6 | 1.8 | 0.77 | 0.23 |

The chemical states of all the other constituents subjected to the XPS experiment showed a normal presence of oxide species for niobium, vanadium, titania, and palladium as well as the presence of metallic gold particles (Figure S1).

Typical TEM images of the bimetallic PdAu-loaded catalysts were acquired and are shown in Figure 4. The TiO$_2$ support (in grey on the figures) appears in the form of aggregated grains on which metal nanoparticles are deposited (dark spots, Figure 4a). Pictures taken at high magnification show that the size of the individual TiO$_2$ grains is of the order of 10–20 nm (Figure 4b) and their crystalline nature (anatase) is confirmed by the plans clearly observed at high resolution (Figure 4c). The coupling of the TEM images' recording with EDS analysis indicates that the big dark spots with diameters varying between 20 and 50 nm correspond predominantly to gold nanoparticles, whereas palladium is much better dispersed, being present all over the support as nanoparticles with sizes below 5 nm (Figure 4a). EDS measurements also confirmed the good dispersion of the promoting agents that were homogeneously detected on the support. For all bimetallic samples, the EDS study showed that the two metals were dispersed separately, in agreement with XPS analysis of the bimetallic samples that proved the existence of metallic gold particles, with no proof of the presence of alloys. Nevertheless, TEM/EDS images of the bimetallic samples revealed the presence of Pd-Au core-shell layouts, palladium forming a shell around the gold nanoparticles. This is in line with the order of metal addition during stepwise samples preparations, gold being added first, followed by palladium impregnation.

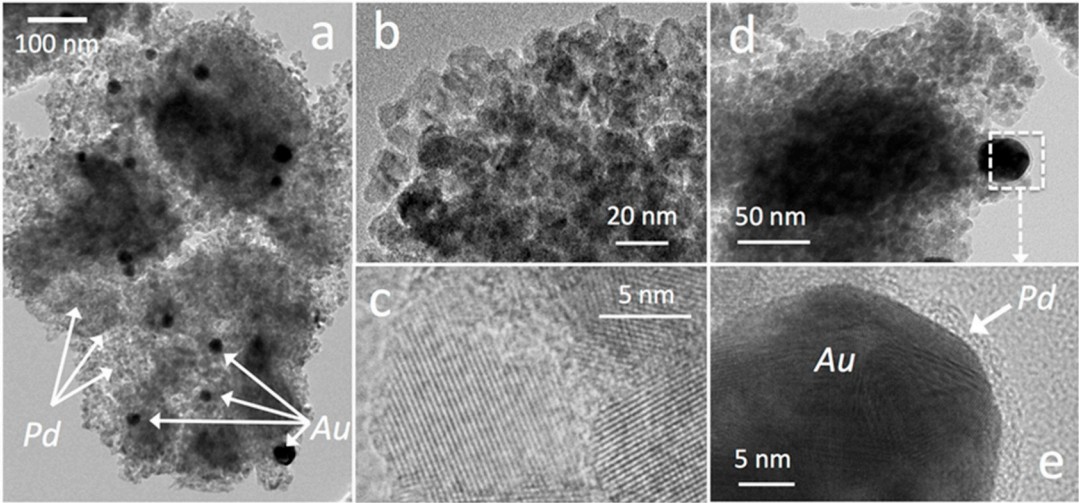

**Figure 4.** Typical TEM images of the PdAu5NbTi bimetallic catalysts showing (**a**) a global view, (**b**) the individual grains of the $TiO_2$ support, (**c**) the anatase crystalline reticular plans, (**d**) an Au nanoparticle, and (**e**) zoomed on the same particle showing an Au core and a Pd shell.

## 3. Discussion

As cited earlier, multiple research teams reported oscillatory behaviors of palladium- or platinum-loaded catalysts in different processes. Some attributed their occurrence to the oxidation of carbonaceous compounds adsorbed at the catalyst surface while others correlated their observations with a change in the oxidation state of palladium [21,26,27,31]. Therefore, this work focused on identifying the implication of both hypotheses in the oscillatory behavior seen in Figure 1. Throughout the investigation, it is important to keep in mind that such a behavior was only observed in the niobium-doped bimetallic sample. Operando DRIFT experiments provided a first insight into causes for the occurrence of oscillations. First, carbonaceous compounds are believed to have adsorbed at the surface of the catalysts. Vibrational bands related to oxygenated compounds reveal even a deposition of such compounds at the surface. These compounds were also identified by mass spectrometry in the exiting gas flow. Similar by-products were isolated and studied by Guisnet et al. [25,32], who characterized coke deposition on oxide catalysts when using noble metal loaded catalysts for the oxidation of VOCs. In fact, the partial oxidation of toluene molecules may generate fragments that can react with the surface or with other fragments adsorbed at the surface of the catalyst, generating in their turn other aliphatic or oxygenated compounds. Indeed, Lee et al. [23] and Tsou et al. [26,27] claim that coke molecules were the main reason behind their observed oscillations. Lee et al. explain that the cycle-like performance was caused by the deposition of coke molecules on the NaX zeolite and their subsequent removal by oxygen, thus inducing a regeneration of the catalyst [23]. Tsou et al. also state that the oxidation process of coke at the surface of the catalyst and a release of transformed molecules from the surface to the exit stream [26]. Moreover, they claim that two types of coke, an active type and a less active one, were observed. In a proposed mechanism, they detail that when the active coke is consumed and transformed to $CO_2$, the reaction temperature returns to its standard value and carbon dioxide yields decrease, thus causing a sharp decrease in conversion rates. This effect is reproduced when active coke species are adsorbed again onto the catalyst. Thus, the deposition of coke shown in Figure 2 induces the production of higher amounts of $CO_2$ and can possibly increase the reaction temperature, thus also boosting the oxidation of VOC molecules. In this case, the formation and consecutive adsorption and desorption of such compounds occurring throughout the ageing test has a direct link with the oscillatory performance of the PdAu5NbTi catalyst.

Surface OH radicals present at the surface of the catalyst also intervene in inducing variations in the catalyst's performance. In fact, variations in the intensities of vibrational bands detected at wavenumber values greater than 3100 $cm^{-1}$ are correlated with variations in the presence of OH

radicals at the surface of the PdAu5NbTi catalyst. Decreases are correlated with a consumption of OH radicals during the reaction and, consequently, with increases in the conversion yields whereas increases are correlated with a return to ~15% of conversion and indicate a possible regeneration of these radicals at the catalyst surface. Such variations are caused by consecutive reduction—oxidation reactions of the active phase at the surface of the catalyst. In fact, Bychkov et al. [19–21] correlated the oscillating effect with variations in surface oxygen amounts at the surface of their $Pd/Al_2O_3$ catalyst. They consequently stated that these oscillations are due to the redox changes of the active phase constituent, in this case, palladium. Since variations of the presence of OH radicals were observed at the surface of the V-doped bimetallic catalyst, but were not accompanied by an oscillatory behavior, it is our belief that oscillations in this case are directly linked to the doping of the bimetallic catalyst with niobium.

Going back to the literature, Tauster et al. [33] claim that the presence of niobium in an inorganic matrix generates interactions in the form of an electron transfer caused by an easy reduction of niobium species. Ruiz-Puigdollers et al. [34] state that doping a metal oxide, $M_xO_y$, with a heteroatom possessing a different valence than the metal, M, results in a charge imbalance by the partial reduction of the dopant. The charge imbalance is thus compensated by the creation of defects in the structure, in other words, by ejecting oxygen atoms from the matrix. One can imagine that by doing so, niobium doped samples would show a higher share of mobile oxygen species compared to other catalysts, which is indeed what XPS measurements showed (Table 1). Therefore, titania doping with niobium enhances the presence of mobile oxygen species at the surface of the catalyst. According to Ruiz-Puigdollers et al. [34], nanostructuring of a metal oxide also enhances the creation of oxygen deficient sites. TEM experiments showed that palladium nanoparticles occupied the surface of the Nb-doped catalyst. Moreover, in a previous work, Pd species present in niobium-doped samples were seen as more easily reducible than those in V-doped samples [24]. Furthermore, adding gold to the Nb-doped catalyst increased the presence of oxygen vacancies (Table 1). Indeed, gold and palladium particles at the surface of the niobium-doped sample are present as separate particles or in a core-shell morphology (Figure 4). Works by Hosseini et al. [11–13] argue that a core-shell layout provides an increase in the bimetallic catalyst's performance towards the oxidation of VOCs. In fact, such a proximity between both types of particles may increase the synergy between (i) both constituents of the active phase, and (ii) between the active phase and the supporting material. Such a synergy would enhance the redox behavior of the active phase, a necessary behavior for a high performance in the oxidation of VOCs. The combination of greater oxygen defects and easiness in the reducibility of the active phase results in the cyclic behavior of the catalyst's performance at low conversion rates.

Based on the preceding discussion and the experimental results presented in this work, a reaction scheme proposed in Figure 5 highlights the behavior of the PdAu5NbTi catalyst under ageing test conditions. Initially, PdO and metallic Au particles are present at the surface of the catalyst. A reduction of PdO occurs through the oxidation of organic molecules. Total and partial oxidation of toluene and by-products occurs at the surface of Pd, Au, and the $TiO_2$ material. Partially burnt molecules adsorb at the surface of the catalyst and react either with oxygen species ($O_I$ and $O_{II}$) or with other organic fragments adsorbed at the same surface. A supplement of $O_{II}$ species is provided mainly by the redox behavior of Au and Pd particles, but also by oxygen species chemisorbed at defect sites on the surface of the catalyst caused by the partial reduction of $Nb^{5+}$. Meanwhile, some coke molecules formed and adsorbed at the surface may resist oxidation and cause a carbonization of reduced Pd particles. A carbonization of Pd(0) species was investigated by Bychkov et al. [19–21] through Thermal Gravimetric Analysis-Mass Spectrometry (TGA-MS) experiments where they reported alternative increases and decreases in the weight of a metallic Pd-based material after contact with coke substances in a highly oxidative atmosphere. They stated that these coke molecules adsorb onto Pd particles temporarily before being evacuated by excess oxygen molecules. Carbonization continues until an excess of $O_{II}$ species and $O_2$ molecules provided by the air flow build-up on the surface of the catalyst. An oxidation of coke molecules then occurs and results in the release of $CO_2$, $H_2O$, and other

physisorbed organic molecules, a re-oxidation of palladium particles by excess surface oxygen, and a boost in the performance of the PdAu5NbTi catalyst.

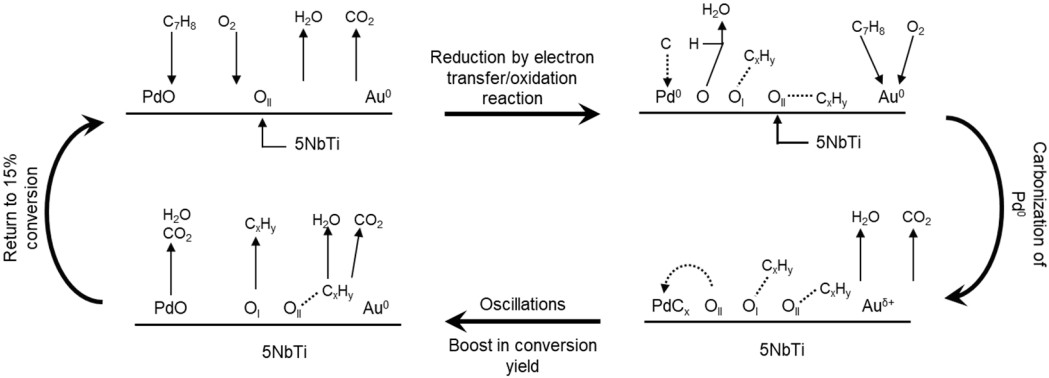

**Figure 5.** Proposed reaction scheme for the degradation of toluene over the PdAu5NbTi catalyst under ageing conditions.

## 4. Materials and Methods

### 4.1. Preparation of Catalysts

After treatment of the synthesized supports at 400 °C, noble metal loaded bimetallic catalysts were synthesized by the consecutive application of the Deposition-Precipitation (DP) and aqueous Impregnation (IMP) techniques. DP of 1 wt % Au was achieved using $HAuCl_4 \cdot 3H_2O$ (Sigma-Aldrich, Saint Quentin Fallavier, France) as the Au precursor. The required amount of doped-$TiO_2$ was dispersed into a solution containing $HAuCl_4$, which was then heated to 80 °C. The solution's pH was adjusted to 8 by dropwise addition of NaOH. The mixture was stirred at 80 °C for 4 h. After DP, samples were washed with water (60 °C) six times, and dried at 80 °C for 24 h. Then, IMP of 0.5 wt % Pd was conducted using $Pd(NO_3)_2$ (Alfa Aesar, Karlsruhe, Germany) as the Pd precursor. The as-prepared catalysts were dried at 80 °C for 24 h, and all samples were calcined at 400 °C (1 °C $min^{-1}$) under flowing air (2 L $h^{-1}$) for 4 h. This method ensures the synthesis of bimetallic Pd-Au-loaded catalysts containing some Pd(shell)-Au(core) morphologies [13], similar to those synthesized and tested by Enache et al. [9].

### 4.2. Catalyst Characterization

Noble metal content of prepared catalysts was determined through elemental analysis performed at the French National Scientific Research Centre (CNRS–Vernaison) using an inductively coupled plasma optical emission spectroscopy and mass spectroscopy (ICP/OES/MS) (Thermo Fischer Scintific iCAP 7000 Plus, Waltham, Ma, USA) after digesting samples in an HF and HCl mixture.

The morphologies of the $TiO_2$ grains and the size and location of the metal particles constituting the active phase (Pd and Au) were determined by transmission electron microscopy on a 2010 JEOL TEM microscope (Tokyo, Japan) operating at 200 keV (LaB6 gun) equipped with an energy dispersive spectroscopy (EDS) probe for local chemical analyses. Before observations, the samples were deposited on a cupper grid covered with a film of carbon and was gently shaken to avoid big deposits.

The specific surface areas ($S_{BET}$) of the catalysts were measured by the BET method with an Ankersmit Surface Area Analyzer (Villeneuve d'Asq, France). The adsorption of a 30% $N_2$ + 70% He mixture was carried out at −196 °C. Rapid heating of the sample resulted in desorption of the gaseous nitrogen that was then quantized with a thermal conductivity detector (TCD).

XPS experiments were performed using an AXIS Ultra DLD Kratos spectrometer (Kratos, Manchester, UK) equipped with a monochromatised aluminium source (Al Kα = 1486.7 eV) and charge compensation gun. All binding energies were referenced to the C 1s core level at 285 eV. Simulation of the experimental photo-peaks was carried out using a mixed Gaussian/Lorentzian peak

fit procedure according to the software supplied by CasaXPS. Semi-quantitative analysis accounted for a nonlinear Shirley background subtraction. The XPS quantification was performed from the study of peak core levels Au 4f, Pd 3d, Nb 3d, V 2p, Ti 2p, C 1s, and O 1s.

### 4.3. Catalytic Activity

The activity of each catalyst (100 mg) was determined by measuring the conversion of toluene (1000 ppm) in flowing air (flow rate = 100 mL min$^{-1}$). The oxidation reaction was carried out in a fixed-bed microreactor operating between 25 and 400 °C (1 °C min$^{-1}$). Prior to the oxidation reactions, the catalysts were calcined at 400 °C (1 °C min$^{-1}$) in flowing air (2 L h$^{-1}$), and then reduced in flowing hydrogen (2 L h$^{-1}$) at 200 °C (1 °C min$^{-1}$).

The 100 h ageing of each catalyst was measured in a constant flow of a mixture of air and toluene (1000 ppm) under the same conditions as the catalytic test, and was followed by Operando Diffuse Reflectance Infrared Fourier Transform (DRIFT) spectroscopy and mass spectrometry. The Operando DRIFT spectroscopy was performed in the 1000 and 4000 cm$^{-1}$ range using an Equinox 55 Bruker spectrometer (Billerica, MA, USA) equipped with a liquid nitrogen cooled mercury-cadmium-telluride (MCT) detector. Mass spectrometry of the exiting gas flow was performed using a Pfeiffer-Vacuum Omnistar™ Quadrupole Mass spectrometer (QMS) 205 (Pfeiffer Vacuum, Asslar, Karlsruhe, Germany).

## 5. Conclusions

Bimetallic Pd-Au supported doped TiO$_2$ catalysts were investigated in ageing tests to underline their stability under extreme testing conditions. Both catalysts seemed to maintain a good activity at low conversion rates even after being exposed to 110 h of a gaseous toluene/air stream. However, the PdAu5NbTi sample showed a significantly different behaviour than its counterpart, the PdAu5VTi sample. A cyclic-like performance was observed for the Nb-doped catalyst, and through operando and XPS measurements, these oscillations can be partially attributed to the participation of surface oxygen, coke molecules, and the redox behaviour of Pd particles. The desorption of adsorbed coke and VOC molecules and their oxidation in the reaction chamber participates not only in increasing CO$_2$ yields, but also in boosting the catalysts activity. This was verified through IR results, which showed a decrease in surface bonds of the catalyst, meaning a possible desorption of the carbonaceous compounds arising from incomplete combustion that are adsorbed on the surface of the tested material. By combining the operando DRIFT results obtained for PdAu5NbTi with XPS results, we can conclude that interactions between the support and the active phase have a direct effect on the regeneration of the catalyst's surface under ageing. A charging of gold and palladium species affects their coordination number, which in turn favors the elimination of weakly adsorbed pollutant molecules. This causes a brief boost in the oxidation process followed by a return to normality, which explains the fluctuating performance of the catalyst. However, the type of carbonaceous compounds triggering these fluctuations has not been yet investigated. A quantitative and qualitative investigation of the adsorbed VOC and coke molecules is therefore crucial as it provides a greater knowledge of what type of carbonaceous compound triggers these fluctuations.

**Supplementary Materials:** The following are available online at http://www.mdpi.com/2073-4344/8/12/574/s1, Figure S1. Operando DRIFT results of the PdAu5VTi catalyst; Figure S2. XPS spectra of mono and bimetallic Nb- (a, c and e) and V-doped (b, d and f) TiO$_2$ samples; Table S1. BET surface area measurements, noble metal content and T50 values for the oxidation of toluene of bimetallic loaded doped catalysts compared to a previously tested PdAu/TiO$_2$ sample; Table S2. Observed DRIFT absorption bands and their assignments for the PdAu5NbTi and PdAu5VTi catalysts as seen in Figure 2 and Figure S1 respectively; Table S3. Binding energy and quantitative analysis values of XPS experiments conducted on mono and bimetallic Nb- and V-doped catalysts.

**Author Contributions:** T.B. prepared the materials, conducted the experiments and wrote the paper. J.C.R. helped the preparation of materials and revised the paper. D.C., J.-F.L., J.-M.G. conducted the XPS analysis. S.C. and P.M. conducted the TEM analysis. B.-L.S., R.C. and S.S. supervised the work. All authors contributed to the data interpretation and discussion.

**Funding:** The authors thank the European community through the Interreg IV REDUGAZ (No. FW 4.1.5) and Interreg V DEPOLLUTAIR (No. 1.1.18) projects for financial supports (European Regional Development Fund).

**Acknowledgments:** The authors thank the European community through the Interreg IV REDUGAZ (No. FW 4.1.5) and Interreg V DEPOLLUTAIR (No. 1.1.18) projects.

**Conflicts of Interest:** The authors declare no conflict of interest.

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
