# Peer review of "Oscillatory Behavior of Pd-Au Catalysts in Toluene Total Oxidation"

_catalysts, doi:10.3390/catal8120574_

Round 1

Reviewer 1 Report

The paper "Oscillatory behavior of Pd-Au catalysts in toluene total oxidation " (Manuscript Number: catalysts-389641) is devoted to the study of the activity of bimetallic Pd-Au doped hierarchically structured titania catalysts in the toluene oxidation.

The paper is well-written. The study of catalyst structure and catalytic activity is well described.

I include one minor comment:

The abbreviation “VOC” has to be deciphered in the abstract (row 18).

I think that the manuscript may be published after a minor improvement.

Author Response

Madam, Sir,

First, we would like to extend our gratitude for your reviewing of our work. As advised, we have included the terminology of VOC in the manuscript, line number 36.

Best regards.

Reviewer 2 Report

This works shows the oscillatory behavior of Nb/PdAu catalysts in toluene total oxidation. An explanation of the reasons after this anomalous oscilatory behaviour is attempted by the authors. The aim of the work is significant although there are several points that authors should clarify:

- The explanation for the oscillatory behaviour proposed by the authors is highly tentative. It can make sense but there are not clear evidences. 

- The characterization of used catalysts after specific times on lines or, better, operando analyses are the best options to understand this behaviour. Accordingly a FTIR operando study is conducted. However the quality of the graph is very bad. I can hardly appreciate the shift in the band at ca. 3000 cm-1 or the increase in the intensity of the bands at 1000-2000 cm-1. As this is very important for the interpretation of the results, the quality of Figure 2 should be largely improved.

- Have you repeated the experiment of Figure 2 on Nb-PdAu catalyst and seen the same oscillatory trend? Is this performance reproducible?

- Do you know why this oscillatory behaviour takes place with the Nb-catalyst but not in the V-catalyst?

- What catalysts are more active AuPd or Nb-AuPd/V-AuPd? Is it worthy to add Nb or V to the AuPd-catalyst?

- The TEM study is very poor. For example, I do not know what catalyst has been represented in Figure 4.

Author Response

Madam, Sir,

First, we would like to extend our gratitude for your reviewing of our work. Your comments have been fully taken into consideration and accordingly, you may find our answers for each of your comments listed hereafter:

       I.          The explanation for the oscillatory behaviour proposed by the authors is highly tentative. It can make sense but there are not clear evidences.

The catalytic testing procedure under 100 hours of constant flow has been tested for reproducibility and provided the same result for each of the samples discussed in this paper. It was clear that the oscillatory behaviour was only evidenced after doping with niobium. As explained throughout the manuscript, such oscillations have been witnessed and discussed in the literature. However, none lasted as long as those presented in this work. Our main objective was thus focused on finding answers to how and what may be causing these oscillations. The evidence to our results therefore resides in both the literature and our experimental results. The presence of a higher amount of mobile oxygen was evidenced by XPS. The importance of this presence is thoroughly discussed in the literature and has been directly related to increases in the performances of tested catalysts. Interactions between all four constituents of our catalysts induce two effects: oxygen vacancies and easily reduced palladium particles. These findings have also been discussed accordingly in the literature. Moreover, evidences of these interactions have been highlighted (i) in this manuscript through TEM/EDS measurements, where intimate contact between all elements has been observed and (ii) in a previously published work where H2-TPR experiments showed that nearly all palladium particles have been reduced at low temperatures in niobium-doped samples.[1] These results combined with quantitative calculations of surface oxygen species at the surface of catalysts are in line with the literature’s explanation of the oscillatory behaviour. All the evidence is therefore presented in this work and reinforced by experimental results as well as discussions published in a previous work.

1.            Barakat, T.; Rooke, J.C.; Franco, M.; Cousin, R.; Lamonier, J.-F.; Giraudon, J.-M.; Su, B.-L.; Siffert, S. Pd- and/or Au-Loaded Nb- and V-Doped Macro-Mesoporous TiO2 Supports as Catalysts for the Total Oxidation of VOCs. Eur. J. Inorg. Chem. 2012, 2012, 2812–2818, doi:10.1002/ejic.201101233.

The above mentioned reference leads to our previsouly published work on Nb- and V-doping and has been included in the manuscript, lines 78 and 223.

     II.          The characterization of used catalysts after specific times on lines or, better, operando analyses are the best options to understand this behaviour. Accordingly a FTIR operando study is conducted. However the quality of the graph is very bad. I can hardly appreciate the shift in the band at ca. 3000 cm-1 or the increase in the intensity of the bands at 1000-2000 cm-1. As this is very important for the interpretation of the results, the quality of Figure 2 should be largely improved.

As advised, the resolution of figure 2 has been improved. an additional sub-figure projecting changes observe around 3100 cm-1 has been added along with a more accurate description of the bands enduring changes.

    III.          Have you repeated the experiment of Figure 2 on Nb-PdAu catalyst and seen the same oscillatory trend? Is this performance reproducible?

Ageing tests for PdAu5NbTi and PdAu5VTi samples were repeated twice under the same flowing conditions. The second testing procedure produced the same results as the first testing round. Therefore, the performance observed for both samples and most importantly for the PdAu5NbTi sample is unquestionably reproducible.

    IV.          Do you know why this oscillatory behaviour takes place with the Nb-catalyst but not in the V-catalyst?

In this present work, the observation of a unique oscillatory behaviour on niobium-doped catalysts was investigated as to understand its nature on the atomic level. A comparison between V-doped and Nb-doped catalysts is currently being investigated in order to explain – experimentally – why the oscillatory effect took place only on the Nb-doped samples. The study extends to an investigation of the behaviour of catalysts containing other dopants capable of providing a charge transfer to the metal active phase constituents and possibly, an oscillatory behaviour under the same reaction conditions as described in this work.

      V.          What catalysts are more active AuPd or Nb-AuPd/V-AuPd? Is it worthy to add Nb or V to the AuPd-catalyst?

The performances of AuPd titania-based samples have been evaluated in the oxidation of toluene in earlier works of ours. Doping these samples with niobium or vanadium has proven to being extremely interesting in the total elimination of toluene, again, as part of an earlier published work. The amount of dopants used was 5 at% of the whole matrix, which means that, in the case of a large scale synthesis, a slight increase in the cost of the material can be noted. The greater increase in the performance of the catalyst justifies the necessity and interest in adding Nb or V to the AuPD-catalysts.

    VI.          The TEM study is very poor. For example, I do not know what catalyst has been represented in Figure 4.

The TEM investigation in figure 4 has been reviewed and the name of the tested catalyst has been added to the caption. Only the PdAu5NbTi sample was investigated in this case as the proximity and sizes of gold and palladium particles were of a great interest to the study. As discussed before, the morphology with which PdAu particles exist at the surface of the catalyst has a direct role in directing the electron transfer from the support to the active phase thus inducing the low temperature reduction of Pd particles.

We remain at your disposal for any further information or clarification you feel necessary to avoid misleading authors.

Best regards